# Dynamic Revenue Sharing[*]

**Santiago Balseiro**
Columbia University
New York City, NY
srb2155@columbia.edu

**Max Lin**
Google
New York City, NY
whlin@google.com

**Vahab Mirrokni**
Google
New York City, NY
mirrokni@google.com

**Renato Paes Leme**
Google
New York City, NY
renatoppl@google.com

**Song Zuo**[†]
Tsinghua University
Beijing, China
songzuo.z@gmail.com

## Abstract

Many online platforms act as intermediaries between a seller and a set of buyers. Examples of such settings include online retailers (such as Ebay) selling items on behalf of sellers to buyers, or advertising exchanges (such as AdX) selling pageviews on behalf of publishers to advertisers. In such settings, *revenue sharing* is a central part of running such a marketplace for the intermediary, and fixed-percentage revenue sharing schemes are often used to split the revenue among the platform and the sellers. In particular, such revenue sharing schemes require the platform to (i) take at most a constant fraction $\alpha$ of the revenue from auctions and (ii) pay the seller at least the seller declared opportunity cost $c$ for each item sold. A straightforward way to satisfy the constraints is to set a reserve price at $c/(1-\alpha)$ for each item, but it is not the optimal solution on maximizing the profit of the intermediary.

While previous studies (by Mirrokni and Gomes, and by Niazadeh et al) focused on revenue-sharing schemes in static double auctions, in this paper, we take advantage of the repeated nature of the auctions. In particular, we introduce *dynamic revenue sharing schemes* where we balance the two constraints over different auctions to achieve higher profit and seller revenue. This is directly motivated by the practice of advertising exchanges where the fixed-percentage revenue-share should be met across all auctions and not in each auction. In this paper, we characterize the optimal revenue sharing scheme that satisfies both constraints in expectation. Finally, we empirically evaluate our revenue sharing scheme on real data.

## 1 Introduction

The space of internet advertising can be divided in two large areas: *search ads* and *display ads*. While similar at first glance, they are different both in terms of business constraints in the market as well as algorithmic challenges. A notable difference is that in search ads the auctioneer and the seller are the same party, as the same platform owns the search page and operates the auction. Thus search ads are a one-sided market: the only agents outside the control of the auctioneer are buyers. In display ads, on the other hand, the platform operates the auction but, in most cases, it does not own the pages in

---

[*]We thank Jim Giles, Nitish Korula, Martin Pál, Rita Ren and Balu Sivan for the fruitful discussion and their comments on early versions of this paper. We also thank the anonymous reviewers for their helpful comments. A full version of this paper can be found at https://ssrn.com/abstract=2956715.

[†]The work was done when this author was an intern at Google. This author was supported by the National Basic Research Program of China Grant 2011CBA00300, 2011CBA00301, the Natural Science Foundation of China Grant 61033001, 61361136003, 61303077, 61561146398, a Tsinghua Initiative Scientific Research Grant and a China Youth 1000-talent program.

which the ads are displayed, making the main problem the design of a two-sided market, referred to as *ad exchanges*.

The problem of designing an ad exchange can be decomposed in two parts: the first is to design an *auction*, which will specify how an ad impression will be allocated among different prospective buyers (advertisers) and how they will be charged from it. The second component is a *revenue sharing scheme*, which specifies how the revenue collected from buyers will be split between the seller (the publisher) and the platform. Traditionally the problems of designing an auction and designing a revenue sharing scheme have been merged in a single one called *double auction design*. This was the traditional approach taken by [4], [3] and more recently in the algorithmic work of [2, 5]. The goals in those approaches have been to maximize efficiency in the market, maximize profit of the platform and to characterize when the profit maximizing policy is a simple one.

Those objectives however, do not entirely correspond to actual problem faced by advertising exchanges. Take platform-profit-maximization, for example. The ad-exchange business is a highly competitive environment. A web publisher (seller) can send their ad impressions to a dozen of different exchanges. If an exchange tries to extract all the surplus in the form of profit, web publishers will surely migrate to a less greedy platform. In order to retain their inventory, exchanges must align their incentives with the incentives of those of web publishers.

A good practical solution, which has been adopted by multiple real world platforms, is to declare a *fixed revenue sharing scheme*. The exchange promises it will keep at most an $\alpha$-fraction of profits, where the constant $\alpha$ is typically the outcome of a business negotiation between the exchange and the web publisher. After the fraction is agreed, the objective of the seller and the exchange are aligned. The exchange maximizes profits by maximizing the seller's revenue.

If revenue sharing was the only constraint, the exchange could simply ignore sellers and run an optimal auction among buyers. In practice, however, web-publishers have outside options, typically in the form of reservation contracts, which should be taken into account by the exchange. Reservation contracts are a very traditional form of selling display ads that predates ad exchanges, where buyers and sellers make agreements offline specifying a volume of impressions to be transacted, a price per impression and a penalty for not satisfying the contract. Those agreements are entered in a system (for example Google's Doubleclick for Publishers) that manages reservations on behalf of the publisher. This reservation system determines for each arriving impression the best matching offline contract that impression could be allocated to as well as the cost of not allocating that impression. The cost of not allocating an impression takes into account the potential revenue from allocating to a contract and the probability of paying a penalty for not satisfying the contract.

From our perspective, it is irrelevant how a cost is computed by reservation systems. It is sufficient to assume that for each impression, the publisher has an opportunity cost and it is only willing to sell that particular impression in the exchange if its payout for that impression exceeds the cost. Exchanges therefore, allow the publisher to submit a cost and only sell that impression if they are able to pay the publisher at least the cost per that impression.

We design the following simple auction and revenue sharing scheme that we call the naïve policy:

- seller sends to the exchange an ad impression with cost $c$.
- exchange runs a second price auction with reserve $r \geq c/(1 - \alpha)$.
- if the item is sold the exchange keeps an $\alpha$ fraction of the revenue and sends the remaining $1 - \alpha$ fraction to the seller.

This scheme is pretty simple and intuitive for each participant in the market. It guarantees that if the impression is sold, the revenue will be at least $c/(1 - \alpha)$ and therefore the seller's payout will be at least $c$. So both the minimum payout and revenue sharing constraints are satisfied with probability 1. This scheme has also the advantage of decoupling the auction and the revenue sharing problem. The platform is free to use any auction among the buyers as long as it guarantees that whenever the impression is matched, the revenue extracted from buyers is at least $c/(1 - \alpha)$.

Despite being simple, practical and allowing the exchange to experiment with the auction without worrying about revenue sharing, this mechanism is sub-optimal both in terms of platform profit and publisher payout. The exchange might be willing to accept a revenue share lower than $\alpha$ if this grants more freedom in optimizing the auction and extracting more revenue.

More generally, the exchange might exploit the repeated nature of the auction to improve revenue even further by adjusting the revenue share dynamically based on the bids and the cost. In this setting,

we can think of the revenue share constraints to be enforced on average, i.e., over a sequence of auctions the platform is required to bound by $\alpha$ the ratio of the aggregate profit and the aggregate revenue collected from buyers. This allows the platform to increase the revenue share on certain queries and reduce in others.

In the repeated auctions setting, the exchange is also allowed to treat the minimum cost constraint on aggregate: the payout for the seller needs to be at least as large as the sum of costs of the impressions matched. The exchange can implement this in practice by always paying the seller at least his cost even if the revenue collected from buyers is less than the cost. This would cause the exchange to operate at a loss for some impressions. But this can be advantageous for the exchange on aggregate if it is able to offset these losses by leveraging other queries with larger profit margins.

In this paper, we attempt to characterize the optimal scheme for repeated auctions and measure on data the improvement with respect to the simple revenue sharing scheme discussed above.

Finally, while we discuss the main application of our results in the context of advertising exchanges, our model and results apply to the broad space of platforms that serve as intermediaries between buyers and sellers, and help run many repeated auctions over time. The issue of *dynamic* revenue sharing also arises when Amazon or eBay act as a platform and splits revenues from a sale with the sellers, or when ride-sharing services such as Uber or Lyft split the fare paid by the passenger between the driver and the platform. Uber for example mentions in their website[3] that: "Drivers using the partner app are charged an Uber Fee as a percentage of each trip fare. The Uber Fee varies by city and vehicle type and helps Uber cover costs such as technology, marketing and development of new features within the app."

## 1.1 Our Results and Techniques

We propose different designs of auctions and revenue sharing policies in exchanges and analyze them both theoretically and empirically on data from a major ad exchange. We compare against the naïve policy described above. We compare policies in terms of seller payout, exchange profit and match-rate (number of impressions sold). We note that match-rate is an important metric in practice, since it represents the volume of inventory transacted in the exchange and it is a proxy for the volume of the ad market this particular exchange is able to capture.

For the auction, we restrict our attention to second price auctions with reserve prices, since we aim at using theory as a guide to inform decisions about practical designs that can be implemented in real ad-exchanges. To be implementable in practice the designs need to follow the industry practice of running second-price auctions with reserves. This design will be automatically incentive compatible for buyers. On the seller side, instead of enforcing incentive compatibility, we will assume that impression costs are reported truthfully. Note that the revenue sharing contract guarantees, at least partially, when the constraint binds (which always happens in practice), the goals of the seller and the platform are partially aligned: maximizing profit is the same as maximizing revenue. Thus, sellers have little incentive to misreport their costs. In fact, this is one of the main reason that so many real-world platforms such as Uber adopt fixed revenue sharing contracts. In the ads market, moreover, sellers are also typically viewed as less strategic and reactive agents. Thus, we believe that the latter assumption is not too restrictive in practice.[4]

We will also assume Bayesian priors on buyer's valuations and on seller's costs. For the sake of simplicity, we will start with the assumption that seller costs are constant and show in the full version how to extend our results to the case where costs are sampled from a distribution.

We will focus on the exchange profit as our main objective function. While this paper will take the perspective of the exchange, the policies proposed will also improve seller's payout with respect to the naïve policy. The reason is simple: the naïve policy keeps exactly $\alpha$ fraction of the revenue extracted from buyers as profit. Any policy that keeps at most $\alpha$ and improves profit, should improve revenue extracted from buyers at least at the same rate and hence improve seller's payout.

**Single Period Revenue Sharing.** We first study the case where exchange is required to satisfy the revenue sharing constraint in each period, i.e., for each impression at most an $\alpha$-fraction of the

revenue can be retained as profit. We characterize the optimal policy. We first show that the optimal policy always sets the reserve price above the seller's cost, but not necessarily above $c/(1-\alpha)$. The exchange might voluntarily want to decrease its revenue share if this grants freedom to set lower reserve prices and extract more revenue from buyers.

When the opportunity cost of the seller is low, the optimal policy for the exchange ignores the seller's cost and prices according to the optimal reserve price. When the opportunity cost is high, pricing according to $c/(1-\alpha)$ is again not optimal because demand is inelastic at that price. The exchange internalizes the opportunity cost, prices between $c$ and $c/(1-\alpha)$, and reduces its revenue share if necessary. For intermediate values of the opportunity cost, the exchange is better off employing the naïve policy and pricing according to $c/(1-\alpha)$.

**Multi Period Revenue Sharing.** We then study the case where the revenue share constraint is imposed over the aggregate buyers' payments. We provide intuition on the structure of the optimal policy by first solving a Lagrangian relaxation and then constructing an *asymptotically optimal* heuristic policy (satisfying the original constraints) based on the optimal relaxation solution. In particular, we introduce a Lagrange multiplier for the revenue sharing constraint to get the optimal solution to the Lagrangian relaxation. The optimal revenue sharing policy obtained from the Lagrangian relaxation pays the publisher a convex combination between his cost $c$ and a fraction $(1-\alpha)$ of the revenue obtained from buyers. Depending on the value of the multiplier, the reserve price could be below $c$, exposing the platform to the possibility of operating at a loss in some auctions.

The policy obtained from the Lagrangian relaxation, while intuitive, only satisfies the revenue sharing and cost constraints in expectation. Because this is not feasible for the platform, we discuss heuristic policies that approximate that policy in the limit, but satisfy the constraints surely in aggregate over the $T$ periods. Then we discuss an even stronger policy that satisfies the aggregate constraints for any prefix, i.e., at any given time $t$, the constraints are satisfied in aggregate from time $1$ to $t$.

**Comparative Statics.** We compare the structure of the single period and multi period policies. The first insight is that the optimal multi-period policy uses lower reserve prices therefore matching more queries. The key insight we obtain from the comparison is that multi-period revenue sharing policies are particularly effective when markets are thick, i.e. when a second highest bid is above a rescaled version of the cost often and cost are not too high.

**Empirical Insights.** To complement our theoretical results, we conduct an empirical study simulating our revenue sharing policies on real world data from a major ad exchange. The data comes from bids in a second price auction with reserves (for a single-slot), which is truthful. Our study confirms the effectiveness of the multi period revenue sharing policies and single period revenue sharing policies over the naïve policy. The results are consistent for different values of $\alpha$: the profit lifts of single period revenue sharing policies are $+1.23\% \sim +1.64\%$ and the lifts of multi period revenue sharing policies are roughly $5.5$ to $7$ times larger ($+8.53\% \sim +9.55\%$).

We do an extended overview in Section 7, but leave the further details to the full version. We omit the related work here, which can be can be found in the full version.

## 2  Preliminaries

**Setting.** We study a discrete-time finite horizon setting in which items arrive sequentially to an intermediary. We index the sequence of items by $t = 1, \ldots, T$. There are multiple buyers bidding in the intermediary (the exchange) and the intermediary determines the winning bidder via a second price auction. We assume that the bids from the buyers are drawn independently and identically distributed across auctions, but potentially correlated across buyers for a given auction.

We will assume that the profit function of the joint distribution of bids is quasi-concave. The expected profit function corresponds to the expected revenue of a second price auction with reserve price $r$ and opportunity cost $c$:

$$\Pi(r,c) = \mathbb{E}\left[\mathbf{1}\{b^{\mathsf{f}} \geq r\}\left(\max(r, b^{\mathsf{s}}) - c\right)\right].$$

where $b_t^{\mathsf{f}}$ and $b_t^{\mathsf{s}}$ are the highest- and second-highest bid at time $t$. Our assumption on the bid distribution will be as follows:

**Assumption 2.1.** *The expected profit function $\Pi(r,c)$ is quasi-concave in $r$ for each $c$.*

The previous assumption is satisfied, for example, if bids are independent and identically distributed according to a distribution with increasing hazard rates (see, e.g., **(author?)** [1]).

**Mechanism.** The seller submitting the items sets an opportunity cost of $c \geq 0$ for the items. The profit of the intermediary is the difference between the revenue collected from the buyers and the payments made to the seller. The intermediary has agreed to a *revenue sharing scheme* that limits the profit of the intermediary to at most $\alpha \in (0,1)$ of the total revenue collected from the buyers.

The intermediary implements a non-anticipative adaptive policy $\pi$ that maps the history at time $t$ to a reserve price $r_t^\pi \in \mathbb{R}_+$ for the second price auction and a payment function $p_t^\pi : \mathbb{R}_+ \to \mathbb{R}_+$ that determines the amount to be paid to the seller as a function of the buyers' payments. That is, the item is sold whenever the highest bid is above the reserve price, or equivalently $b_t^\mathsf{f} \geq r_t^\pi$. The intermediary's revenue is equal to the buyers' payments of $\max(r_t^\pi, b_t^\mathsf{s})$ and the seller's revenue is given by $p_t^\pi (\max(r_t^\pi, b_t^\mathsf{s}))$. The intermediary's profit is given by the difference of the buyers' payments and the payments to the seller, i.e., $\max(r_t^\pi, b_t^\mathsf{s}) - p_t^\pi (\max(r_t^\pi, b_t^\mathsf{s}))$. From the perspective of the buyers, the mechanism implemented by the intermediary is a second price auction with (potentially dynamic) reserve price $r_t^\pi$. The intermediary's problem amounts to maximizing profits subject to the revenue sharing constraint. The revenue sharing constraint can be imposed at every single period or over multiple periods. We discuss each model at a time.

**Naïve revenue sharing scheme.** The most straightforward revenue sharing scheme is the one that sets a reserve above $c/(1-\alpha)$ and pay the sellers a $(1-\alpha)$-fraction of the revenue:

$$r_t^\pi \geq \tfrac{c}{1-\alpha}, \quad p_t^\pi(x) = (1-\alpha)x. \tag{1}$$

Since the revenue sharing is fixed, the intermediary's profit is given by $\alpha \max(r_t^\pi, b_t^\mathsf{s})$. Thus, the intermediary optimizes profits by optimizing revenues, and the optimal reserve price is given by:

$$r^* = \arg\max_{r \geq c/(1-\alpha)} \Pi(r, 0).$$

The naïve revenue sharing scheme sets a reserve above $c/(1-\alpha)$ and pays the seller $(1-\alpha)$ of the buyers' payments. This guarantees that the payment to the seller is always no less than $c$, by construction, because the payment of the buyers is at least the reserve price. Since the intermediary's profit is a fraction $\alpha$ of the buyers' payment, the seller's cost does not appear in the objective, and the objective of the seller is $\alpha\Pi(r, 0)$. Note, however, that the seller's cost does appear as a constraint in the intermediary's optimization problem: the reserve price should be at least $c/(1-\alpha)$.

This is the baseline that we will use to compare the proposed policies with in the experiment section. This policy is suboptimal for various reasons. Consider for example the extreme case where the buyers alway bid more than $c$ and less than $c/(1-\alpha)$. In this case, the profit from the naïve revenue sharing scheme is zero. However, the intermediary can still obtain a non-zero profit by setting the reserve somewhere between $c$ and $c/(1-\alpha)$, which results in a revenue share less than $\alpha$. If the revenue sharing constraint is imposed over multiple periods instead of each single period, we are able to dynamically balance out the deficit and surplus of the revenue sharing constraint over time.

## 3 Single Period Revenue Sharing Scheme

In this case the revenue sharing scheme imposes that in every single period the profit of the intermediary is at most $\alpha$ of the buyers' payment. We start by formulating the profit maximization problem faced by the intermediary as a mathematical program with optimal value $J^S$.

$$J^S \triangleq \max_\pi \; \sum_{t=1}^T \mathbb{E}\left[ \mathbf{1}\{b_t^\mathsf{f} \geq r_t^\pi\} \left(\max(r_t^\pi, b_t^\mathsf{s}) - p_t^\pi (\max(r_t^\pi, b_t^\mathsf{s}))\right)\right] \tag{2a}$$

$$\text{s.t. } p_t^\pi(x) \geq (1-\alpha)x, \quad \forall x \tag{2b}$$

$$p_t^\pi(x) \geq c, \quad \forall x. \tag{2c}$$

The objective (2a) gives the profit of the intermediary as the difference between the payments collected from the buyers and the payments made to the seller. The revenue sharing constraint (2b) imposes that intermediary's profit is at most a fraction $\alpha$ of the total revenue, or equivalently $(x - p_t^\pi(x))/x \leq \alpha$ where $x$ is the payment from the buyers. The floor constraint (2c) imposes that the seller is paid at least $c$. These constraints are imposed at every auction.

We next characterize the optimal decisions of the seller in the single period model. Some definitions are in order. Let $r^*(c)$ be an optimal reserve price in the second price auction if the seller's cost is $c$:

$$r^*(c) = \arg\max_{r \geq 0} \Pi(r, c).$$

To avoid trivialities we assume that the optimal reserve price is unique. Because the profit function $\Pi(r, c)$ has increasing differences in $(r, c)$ then the optimal reserve price is non-decreasing with the cost, that is, $r^*(c) \geq r^*(c')$ for $c \geq c'$.

Our main result in this section characterizes the optimal decision of the intermediary in this model.

**Theorem 3.1.** *The optimal decision of the intermediary is to set $p_t^\pi(x) = \max(c, (1-\alpha)x)$ and $r_t^\pi = \max\{\min\{\bar{c}, r^*(c)\}, r^*(0)\}$ where $\bar{c} = c/(1-\alpha)$.*

The reserve price $\bar{c} = c/(1-\alpha)$ in the above theorem is the naïve reserve price that satisfies the revenue sharing scheme by inflating the opportunity cost by $1/(1-\alpha)$. When the opportunity cost $c$ is very low ($\bar{c} \leq r^*(0)$), pricing according to $\bar{c}$ is not optimal because demand is elastic at $\bar{c}$ and the intermediary can improve profits by increasing the reserve price. Here the intermediary ignores the opportunity cost, prices optimally according to $r_t^\pi = r^*(0)$ and pays the seller according to $p_t^\pi(x) = (1-\alpha)x$. When the opportunity cost $c$ is very high ($\bar{c} \geq r^*(c)$), pricing according to $\bar{c}$ is again not optimal because demand is inelastic at $\bar{c}$ and the intermediary can improve profits by decreasing the reserve price. Here the intermediary internalizes the opportunity cost, prices optimally according to $r_t^\pi = r^*(c)$ and pays the seller according to $p_t^\pi(x) = \max(c, (1-\alpha)x)$.

## 4   Multi Period Revenue Sharing Scheme

In this case the revenue sharing scheme imposes that the aggregate profit of the intermediary is at most $\alpha$ of the buyers' aggregate payment. Additionally, in this model the opportunity costs are satisfied on an aggregate fashion over all actions, that is, the payments to the seller need to be at least the floor price times the number of items sold. The intermediary decision's problem can be characterized by the following mathematical program with optimal value $J^M$, where $x_t^\pi = \max(r_t^\pi, b_t^s)$

$$J^M \triangleq \max_\pi \; \sum_{t=1}^T \mathbb{E}\left[\mathbf{1}\{b_t^f \geq r_t^\pi\}(x_t^\pi - p_t^\pi(x_t^\pi))\right] \tag{3a}$$

$$\text{s.t.} \; \sum_{t=1}^T \mathbf{1}\{b_t^f \geq r_t^\pi\}(p_t^\pi(x_t^\pi) - (1-\alpha)x_t^\pi) \geq 0, \tag{3b}$$

$$\sum_{t=1}^T \mathbf{1}\{b_t^f \geq r_t^\pi\}(p_t^\pi(x_t^\pi) - c) \geq 0, . \tag{3c}$$

The objective (3a) gives the profit of the intermediary as the difference between the payments collected from the buyers and the payments made to the seller. The revenue sharing constraint (3b) imposes that intermediary's profit is at most a fraction $\alpha$ of the total revenue. The floor constraint (3c) imposes that the seller is paid at least $c$. These constraints are imposed over the whole horizon.

The stochastic decision problem (3) can be solved via Dynamic Programming. To provide some intuition of the structure of the optimal solution we solve a Lagrangian relaxation of the problem where we introduce a dual variable $\lambda \geq 0$ for the floor constraint (3c) and a dual variable $\mu \geq 0$ for the revenue sharing constraint (3b). Lagrangian relaxations provide upper bounds on the optimal objective value and introduce heuristic policies of provably good performance in many settings (e.g., see [7]). Moreover, we shall see the optimal policy derived from the Lagrangian relaxation is optimal for problem (3) if constraints (3c) and (3b) are imposed in expectation instead of almost surely:

**Theorem 4.1.** *Let $\mu^* \in \arg\min_{0 \leq \mu \leq 1} \hat{\phi}(\mu)$. The policy $p_t^\pi(x) = (1-\mu^*)c + \mu^*(1-\alpha)x$ and $r_t^\pi = r^*(c(\mu^*))$ is optimal for problem* (3) *when constraints* (3c) *and* (3b) *are imposed in expectation instead of almost surely, where*

$$\hat{\phi}(\mu) \triangleq T\big(1 - \mu(1-\alpha)\big) \sup_r \Pi\big(r, \tfrac{(1-\mu)c}{1-\mu(1-\alpha)}\big).$$

**Remark 4.2.** *Although the multi period policy proposed is not a solution to the original program* (3)*, we emphasize that it naturally induces heuristic policies (e.g., see Algorithm 1) that are asymptotically optimal solutions to the original multi period problem* (3) *without relaxation (see Theorem 6.1).*

## 5   Comparative Analysis

We first compare the optimal reserve price of the single period and multi period model.

**Proposition 5.1.** *Let $r^S \triangleq \max\{\min\{\bar{c}, r^*(c)\}, r^*(0)\}$ be the optimal reserve price of the single period constrained model and $r^M \triangleq r^*(c(\mu^*))$ be the optimal reserve price of the multi period constrained model. Then $r^S \geq r^M$.*

The previous result shows that the reserve price of the single-period constrained model is larger or equal than the one of the multi-period constrained model. As a consequence, in the multi-period constrained model items are allocated more frequently and the social welfare is larger.

We next compare the intermediary's optimal profit under the single period and multi period model. This result quantifies the benefits of dynamic revenue sharing and provides insight into when dynamic revenue sharing is profitable for the intermediary.

**Proposition 5.2.** *Let $\mu^S \in [0, 1]$ be such that $r^*(c(\mu^S)) = r^S$. Then*

$$J^S \leq J^M \leq J^S + (1 - \mu^S)T\mathbb{E}\left[(1 - \alpha)b^{\mathsf{s}} - c\right]^+.$$

The previous result shows that the benefit of dynamic revenue sharing is driven, to a large extent, by the second-highest bid and the opportunity cost $c$. If the market is thin and the second-highest bid $b^{\mathsf{s}}$ is low, then the truncated expectation $E \triangleq \mathbb{E}\left[(1 - \alpha)b^{\mathsf{s}} - c\right]^+$ is low and the benefit from dynamic revenue sharing is small, that is, $J^S \sim J^M$. If the market is thick and the second-highest bid $b^{\mathsf{s}}$ is high, then the benefit of dynamic revenue sharing depends on the opportunity cost $c$. If the floor price $c$ is very low, then $r^S = r^*(0)$ and $\mu^S = 1$, implying that the coefficient in front of $E$ is zero, and there is no benefit of dynamic revenue sharing $J^S = J^M$. If the floor price $c$ is very high, then $r^S = r^*(c)$ and $\mu^S = 0$, implying that the coefficient in front of $E$ is 1. However, in this case the truncated expectation $E$ is small and again there is little benefit of dynamic revenue sharing, that is, $J^S \sim J^M$. *Thus the sweet spot for dynamic revenue sharing is when the second-highest bid is high and the opportunity cost is neither too high nor too low.*

# 6 Heuristic Revenue Sharing Schemes

So far we focused on the theory of revenue sharing schemes. We now switch our focus to applying insights derived from theory to the practical implementation of revenue sharing schemes. First we note that while the policy in the statement of Theorem 4.1 is only guaranteed to satisfy constraints in expectations, a feasible policy of the stochastic decision problems should satisfy the constraints in an almost sure sense.

We start then by providing two transformations that convert a given policy satisfying constraints in expectation to another policy satisfying the constraints in every sample path.

## 6.1 Multi-period Refund Policy

Our first transformation will keep track of how much each constraint is violated and will issue a refund to the seller in the last period (see Algorithm 1).

---

**ALGORITHM 1:** Heuristic Refund Policy from Lagrangian Relaxation

1: Determine the optimal dual variable $\mu^* \in \arg\min_{0 \leq \mu \leq 1} \hat{\phi}(\mu)$
2: **for** $t = 1, \dots, T$ **do**
3:      Set the reserve price $r_t^\pi = r^*(c(\mu^*))$
4:      **if** item is sold, that is, $b_t^{\mathsf{f}} \geq r_t^\pi$ **then**
5:          Collect the buyers' payment $x_t^\pi = \max(r_t^\pi, b_t^{\mathsf{s}})$
6:          Pay the seller $p_t^\pi(x_t^\pi) = (1 - \mu^*)c + \mu^*(1 - \alpha)x_t^\pi$
7:      **end if**
8: **end for**
9: Let $D^F = \sum_{t=1}^T \mathbf{1}\{b_t^{\mathsf{f}} \geq r_t^\pi\}\left(p_t^\pi(x_t^\pi) - c\right)$ be the floor deficit.
10: Let $D^R = \sum_{t=1}^T \mathbf{1}\{b_t^{\mathsf{f}} \geq r_t^\pi\}\left(p_t^\pi(x_t^\pi) - (1 - \alpha)x_t^\pi\right)$ be the revenue sharing deficit.
11: Pay the seller $-\min\{D^F, D^R, 0\}$

---

The following result analyzes the performance of the heuristic policy. We omit the proof as this is a standard result in the revenue management literature.

**Theorem 6.1** (Theorem 1, [7]). *Let $J^H$ be the expected performance of the heuristic policy. Then*

$$J^H \leq J^M \leq J^H + O(\sqrt{T}).$$

The previous result shows that the heuristic policy given by Algorithm 1 is asymptotically optimal for the multi-period constrained model, that is, it implies that $J^H/J^M \to 1$ as $T \to \infty$. When the

number of auctions is large, by the Law of Large Numbers, stochastic quantities tend to concentrate around their means. So the floor and revenue sharing deficits incurred by violations of the respective constraints are small relative to the platform's profit and the policy becomes asymptotically optimal.

**Prefix and Hybrid Revenue Sharing Policies.** We also propose several other policies satisfying even more stringent business constraints: revenue sharing constraints can be satisfied in aggregate over all past auctions *at every point in time*. Construction details could be found in the full version.

# 7 Overview of Empirical Evaluation

In this section, we use anonymized real bid data from a major ad exchange to evaluate the policies discussed in previous sections. Our goal will be to validate our insights on data. In the theoretical part of this paper we made simplifying assumptions, that not necessarily hold on data. For example, we assume quasi-concavity of the expected profit function $\Pi(r, c)$. Even though this function is not concave, we can still estimate it from data and optimize using linear search. Our theoretical results also assume we have access to distributions of buyers' bids. We build such distributions from past data. Finally, in our real data set bids are not necessarily stationary and identically distributed over time. Even though there might be inaccuracies from bids changing from one day to another, our revenue sharing policies are also robust to such non-stationarity.

**Data Sets** The data set is a collection of auction records, where each record corresponds to a real time auction for an impression and consists of: (i) a seller (publisher) id, (ii) the seller declared opportunity cost, and (iii) a set of bid records. The maximum revenue share $\alpha$ that the intermediary could take is set to be a constant. To show that our results do not rely on the selection of this constant, we run the simulation for different values of $\alpha$ ($\alpha = 0.15, 0.2, 0.25$), while due to the limit of space, we only present the numbers for $\alpha = 0.25$ and refer the readers to the full version for more details.

Our data set will consist of a random sample of auctions from 20 large publishers over the period of 2 days. We will partition the data set in a *training set* consisting of data for the first day and a *testing set* consisting of data for the second day.

**Preprocessing Steps** Before running the simulation, we need to do some preprocessing of the data set. The goal of the preprocessing is to learn the parameters required by the policies we introduced for each seller, in particular, the optimal reserve function $r^*$ and the optimal Lagrange multiplier $\mu^*$. We will do this estimation using the training set, i.e., the data from the first day.

The first problem is to estimate $\Pi(r, c)$ and $r^*(c)$. To estimate $\Pi(r, c)$ for a given impression we look at all impressions in the training set with the same seller and obtain a list of $(b^{\mathsf{f}}, b^{\mathsf{s}})$ pairs. We build the empirical distribution where each of those pairs is picked with equal probability. This allows us to evaluate and optimize $\Pi(r, c)$ with a single pass over the data using the technique described in [6]. For each seller, to estimate $\mu^*$, we enumerate different $\mu$'s from the discretization of $[0, 1]$ (denoted by $D$) and evaluate the profits of these policies on the training set. Then the estimation ($\hat{\mu}^*$) of $\mu^*$ is the $\mu$ that yields the maximum profit on the training set, i.e., $\hat{\mu}^* \triangleq \arg\max_{\mu \in D} \hat{\mathsf{profit}}(\mu)$

## 7.1 Evaluating Revenue Sharing Policies

We will evaluate the different policies discussed in the paper on testing set (day 2 of the data set) using the parameters $\hat{r}^*(c)$ and $\hat{\mu}^*$ learned from the training set during preprocessing. For each revenue sharing policy we evaluate, we will be concerned with the following metrics: *profit* of the exchange, *payout* to the sellers, *match rate* which corresponds the number of impressions allocated, *revenue* extracted from buyers and *buyers values* which is the sum of highest bids over allocated impressions (we assume that buyers report their values truthfully in the second-price auction). In addition, the average *intermediary's revenue share* will be calculated.

The policies evaluated will be the following: `NAIVE`: naïve policy (Section 2), `SINGLE`: single period policy (Section 3), `REFUND`: multi period refund policy (Algorithm 1), `PREFIX` and `HYBRID`.[5] In Table 1, we report the results of the policies described above or $\alpha = 0.25$ (see the full version for more values of $\alpha$). The metrics are reported with respect to the `NAIVE` policy. In other words, the cell in the table corresponding to revenue of policy P is the revenue lift of P with respect to

| policy | profit | payout | match rate | revenue | buyers values | rev. share |
|--------|--------|--------|-----------|---------|---------------|-----------|
| NAIVE  | 0.00%  | 0.00%  | 0.00%     | 0.00%   | 0.00%         | 25.00%    |
| SINGLE | +1.64% | +2.97% | +1.07%    | +2.64%  | +1.39%        | 24.76%    |
| REFUND | +9.55% | +9.57% | +10.71%   | +9.56%  | +9.64%        | 25.00%    |
| PREFIX | −1.00% | +2.16% | −18.51%   | +1.37%  | −2.90%        | 24.41%    |
| HYBRID | +4.61% | +6.90% | +6.74%    | +6.33%  | +4.55%        | 24.60%    |

Table 1: Performance of the policies for $\alpha = 0.25$.

NAIVE: revenue lift(P) = revenue(P)/revenue(NAIVE) − 1. The only metric that is not reported as a percentage lift is the revenue share in the last column: rev share(P) = profit(P)/revenue(P).

**Interpreting Simulation Results**   What conclusions can we draw from the lift numbers? The first conclusion is that even though the theoretical model deviates from practice in a number of different ways (concavity of $\Pi(r, c)$, precise distribution estimates, stationarity of bids), we are still able to improve over the naïve policy. Notice that the naïve policy implements the optimal reserve price subject to a fixed revenue sharing policy. So all the gains from reserve price optimization are already accounted for in our baseline.

We start by observing that even for SINGLE, which is a simple policy, we are able to considerably improve over NAIVE across all performance metrics. This highlights that the observation that "profit and revenue can be improved by reducing the share taken by the exchange" is not only a theoretical possibility, but a reality on real-world data.

Next we compare the lifts of SINGLE, which enforces revenue sharing constraints per impression, versus REFUND, which enforces constraints in aggregate. We can see that the lift is 5.8 times larger for REFUND compared to SINGLE. For $\alpha = 0.25$, the lift[6] for SINGLE is +1.64% while REFUND is +9.55%. This shows the importance of optimizing revenue shares across all auctions instead of per auction. Additionally, we observe that the match rate and buyers values of REFUND are higher than those of SINGLE. This is in agreement with Proposition 5.1: because the reserve price of the single-period constrained model is typically larger than the one of the multi-period constrained model, we expect REFUND to clear more auctions, which in turns leads to higher buyer values.

Finally, we birefly analyze the performance of PREFIX and HYBRID policies. While PREFIX is proposed to guarantee more stringent constraints, it fails to have a positive impact on profit. Instead, with some slight modifications, HYBRID is able to overcome these shortcomings by granting the intermediary more freedom in picking reserve prices. As a result, we obtain a policy that is consistently better than SINGLE. Even though not as good as REFUND in terms of revenue lift, HYBRID satisfied the more stringent constraints that are not necessarily satisfied by REFUND. To sum up, the policies can be ranked as follows in terms of performance:

$$\text{REFUND} \succ \text{HYBRID} \succ \text{SINGLE} \succ \text{NAIVE} \sim \text{PREFIX}.$$

## Footnotes

[3]See https://www.uber.com/info/how-much-do-drivers-with-uber-make/

[4]While in this paper we focus on the dynamic optimization of revenue sharing schemes when agents report truthfully, it is still an interesting avenue of research to study the broader market design question of designing dynamic revenue sharing schemes while taking into account agents' incentives.

[5]The details of policy `PREFIX` and `HYBRID` are omitted here, see the full version for further details.

[6]The reader might ask how to interpret lift numbers. The annual revenue of display advertising exchanges is on the order of billions of dollars. At that scale, 1% lift corresponds to tens of millions of dollars in incremental annual revenue. We emphasize that this lift is in addition to that obtained by reserve price optimization.

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
