[Supplementary Material]

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

It is important to mention that while we derive policies optimizing exchange profit, there are other business constraints and real world objectives that exchanges need to balance. Most notably, exchanges need to offer attractive terms of trade so as to be competitive with respect to other exchanges by, for example, making sure publishers obtain enough revenue. Exchanges are often concerned about match rate, which measures the volume of impressions transacted in the exchange. For that reason, we also measure how those metrics are affected by our policy and show that they improve.

Our final goal is to confirm the insights obtained from theory. We prove that under concavity conditions on the profit function, the performance of multi period policies over single period ones depend on the relation between second highest bids and costs. Our empirical study confirms the existence of a sweet spot for costs: we evaluate the single period and multi period policies after rescaling costs by different factors. When costs are too low or too high, we observe similar

performance for the two policies. Interestingly, the unscaled costs are in the sweet spot where multi-period policies are particularly effective. These empirical observations are consistent with our prediction for different values of $\alpha$.

## 1.2 Related Work

**Double Auctions.** Revenue sharing schemes have been studied in the context of optimal double auctions. Following the seminal work of Myerson [1981], Myerson and Satterthwaite [1983] presented the optimal double auction in two-sided settings. Recently, Deng et al. [2011] studied multidimensional variants of the Bayesian optimal double auctions and present polynomial-time approximation algorithms for the problem. For the non-Bayesian (prior-free) setting, Deshmukh et al. [2002] studied revenue-maximizing double auctions when the auctioneer has no prior knowledge about bids. More recently, Gomes and Mirrokni [2014] studied revenue-maximizing double auctions in the context of advertising exchanges, and generalized the results of Myerson and Satterthwaite [1983] by studying settings in which the platform's objective function is a convex combination of the seller's profit and the platform's profit, and provided a necessary and sufficient condition under which constant sharing schemes indirectly implement the optimal mechanism. Furthermore, Niazadeh et al. [2014] developed an approximately optimal mechanism for constant revenue-sharing double auctions. To the best of our knowledge, none of the above work consider revenue sharing schemes in repeated auctions.

**Exchange Design.** Our work is also related to the broad question of ad exchange design. We refer to Muthukrishnan [2009] for a survey. Mansour et al. [2012] provides an overview of the auction employed by Google's ad exchange. Feldman et al. [2010], Balseiro et al. [2016] study how should the exchange design auctions when advertisers do not acquire impressions directly from the exchange, but instead contract with intermediaries to acquire impressions on their behalf. These papers, however, take a one-sided approach to the display advertising market and do not take into account the presence of revenue sharing schemes for publishers.

**E-commerce Applications.** The results of this paper apply to various online and offline retailers and e-commerce websites like Amazon and Ebay. More specifically, Ebay applies similar revenue-sharing auctions to the ones studied in this paper when it serves as a broker between a set of buyers and a seller. Roughly speaking, Ebay takes a 9% cut on each sale, referred to as *final value fee*, and also fixed fee for listing an item, referred to as an *insertion fee*. They also apply a convex cost function for the fixed fee as the number of listings, and a maximum of $250 for the final value fee. A recent paper by Jain and Wilkens [2012] studies EBay's double auction problem, but their setting is different from this paper as they consider multiple sellers and one buyer, and explore approximately optimal pricing schemes for this setting.

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

## 4.1 Lagrangian Relaxation

The dual function $\phi(\mu, \lambda)$ is given by supremum of the Lagrangian over the feasible set:

$$
\begin{aligned}
\phi(\mu, \lambda) &\triangleq \sup_{\pi} \sum_{t=1}^{T} \mathbb{E}\left[\mathbf{1}\{b_t^{\mathsf{f}} \geq r_t^{\pi}\}\left(x_t^{\pi} - p_t^{\pi}(x_t^{\pi}) + \lambda\big(p_t^{\pi}(x_t^{\pi}) - c\big) + \mu\big(p_t^{\pi}(x_t^{\pi}) - (1-\alpha)x_t^{\pi}\big)\right)\right] \\
&= \sup_{\pi} \sum_{t=1}^{T} \mathbb{E}\left[\mathbf{1}\{b_t^{\mathsf{f}} \geq r_t^{\pi}\}\left((1 - \mu(1-\alpha))x_t^{\pi} - (1 - \lambda - \mu)p_t^{\pi}(x_t^{\pi}) - \lambda c\right)\right] \\
&= T \sup_{r} \mathbb{E}\left[\mathbf{1}\{b^{\mathsf{f}} \geq r\}\left((1 - \mu(1-\alpha))\max(r, b_t^{\mathsf{s}}) - \lambda c\right)\right] + \mathcal{X}_{\{\lambda + \mu = 1\}} \\
&= T(1 - \mu(1-\alpha))\sup_{r} \Pi\left(r, \frac{\lambda c}{1 - \mu(1-\alpha)}\right) + \mathcal{X}_{\{\lambda + \mu = 1\}}
\end{aligned}
$$

where the third equality follows because bids are i.i.d. and thus the problem is separable, and from optimizing over the payment function $p_t^{\pi}$ and noting that the objective is unbounded unless $\lambda + \mu = 1$ and denoting by $\mathcal{X}_A$ the characteristic function which is zero if $A$ is true and $\infty$ otherwise.

Because the Lagrangian is unbounded unless $\lambda + \mu = 1$, then the optimal dual objective is given by

$$
\inf_{\mu \geq 0, \lambda \geq 0 : \mu + \lambda = 1} \phi(\mu, \lambda) = \inf_{0 \leq \mu \leq 1} \phi(\mu, 1 - \mu) = \inf_{0 \leq \mu \leq 1} \hat{\phi}(\mu)
$$

where

$$
\begin{aligned}
\hat{\phi}(\mu) &\triangleq \phi(\mu, 1 - \mu) = T(1 - \mu(1-\alpha))\sup_{r} \Pi\left(r, \frac{(1-\mu)c}{1 - \mu(1-\alpha)}\right) \\
&= T(1 - \mu(1-\alpha))\Pi\left(r^{*}(c(\mu)), c(\mu)\right) .
\end{aligned} \tag{4}
$$

and $c(\mu) = \frac{1-\mu}{1 - \mu(1-\alpha)} \cdot c \leq c$. Because $\hat{\phi}(\mu)$ is the pointwise maximum of linear functions, then $\hat{\phi}(\mu)$ is convex $\mu$ and thus the dual problem is convex. Because the feasible set is compact then the dual problem admits a solution. Furthermore, from weak duality we have that

$$
J^{M} \leq \inf_{0 \leq \mu \leq 1} \hat{\phi}(\mu).
$$

This suggests us to choose $\mu$ minimizing $\phi(\mu)$ and use it to infer the optimal policy from the dual relaxation. The next result proposes a stationary policy for the intermediary that satisfies the floor and revenue sharing constraints in expectation. The proof is deferred to the appendix.

**Theorem 4.1.** *Let $\mu^{*} \in \arg\min_{0 \leq \mu \leq 1} \hat{\phi}(\mu)$. The policy $p_t^{\pi}(x) = (1 - \mu^{*})c + \mu^{*}(1-\alpha)x$ and $r_t^{\pi} = r^{*}(c(\mu^{*}))$ is optimal for problem* (3) *when constraints* (3c) *and* (3b) *are imposed in expectation instead of almost surely.*

**Remark 4.2.** *Although the multi period policy we just proposed is not a solution to the original program* (3)*, we emphasize that it naturally induces heuristic policies (e.g., see Algorithm 1) that are asymptotically optimal solutions to the original multi period problem* (3) *without relaxation (see Theorem 6.1).*

## 4.2 Random Opportunity Costs

In this section, we extend our characterization result for the multi stage case (Theorem 4.1) to the case where the opportunity cost $c$ for the seller is not fixed but drawn from some distribution for each item $t$. This setting describes the scenario where the sequentially arriving items are heterogeneous and randomly drawn from a population. In practice, impressions are sometimes heterogeneous and the publisher-declared opportunity costs can vary with the attributes of the impressions.

Denote by $c_t$ the opportunity cost at time $t$. The optimization program of the intermediary is as in (3) with the exception that the floor constraint (3c) is now

$$
\sum_{t=1}^{T} \mathbf{1}\{b_t^{\mathsf{f}} \geq r_t^{\pi}\}\left(p_t^{\pi}(x_t^{\pi}) - c_t\right) \geq 0, \tag{5}
$$

and in the objective the expectations are taken over the buyers' bids and the opportunity costs $c_t$.

Because the intermediary observes the opportunity cost declared by the publisher, it can adjust the reserve price for the auction depending on the opportunity cost. Thus, in the Lagrangian relaxation the intermediary can optimize the reserve price point-wise for each opportunity cost and we obtain the dual function:

$$\hat{\varphi}(\mu) = T\big(1 - \mu(1 - \alpha)\big)\mathbb{E}_c\left[\Pi\left(r^*(c(\mu)), c(\mu)\right)\right],$$

where $c(\mu) = \frac{1-\mu}{1-\mu(1-\alpha)} \cdot c$ as before.

**Theorem 4.3.** *Let* $\mu^* \in \arg\min_{0 \leq \mu \leq 1} \hat{\varphi}(\mu)$. *The policy* $p_t^\pi(x) = (1 - \mu^*)c_t + \mu^*(1 - \alpha)x$ *and* $r_t^\pi = r^*(c_t(\mu^*))$ *with* $c_t(\mu) = \frac{1-\mu}{1-\mu(1-\alpha)} \cdot c_t$ *is optimal for problem* (3) *when constraints* (5) *

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

Now we look at how to transform a policy to satisfy even more stringent business constraints. A business constraint that arises in practice is that revenue sharing constraints can be satisfied in aggregate over all past auctions *at every point in time*. Formally, it means that for every prefix of the sequence, we should have:

$$\forall t \geq 1, \sum_{\tau=1}^{t} \mathbf{1}\{b_\tau^f \geq r_\tau^\pi\} \, (p_\tau^\pi(x_\tau^\pi) - (1 - \alpha)x_\tau^\pi) \geq 0. \tag{6}$$

Another important business constraint in practice is that we should pay the seller at least his cost $c_t$ for each impression matched, i.e.:

$$\forall t \geq 1, \mathbf{1}\{b_t^f \geq r_t^\pi\} \, (p_t^\pi(x_t^\pi) - c) \geq 0. \tag{7}$$

For any given revenue sharing scheme $\pi$, we can construct another $\hat{\pi}$ that satisfies the two constraints above and only differs with $\pi$ on the payment rule.

The construction is based on the following ideas: The revenue share constraints (6) are imposed on each prefix of the sequence of the auctions, hence we need to increase the payment to the seller whenever we are about to violate the constraint by simply following the given revenue sharing scheme $\pi$. To do this, we use a "bank account" $B$ to keep the track of the left-hand-side of (6), and make sure that for each period, $B_t = B_{t-1} + p_t^{\hat{\pi}}(x_t^\pi) - (1 - \alpha)x_t^\pi \geq 0$.

The opportunity cost constraints (7) are imposed on each single period, hence we need to increase the payment to the seller if $p_t^\pi(x_t^\pi) < c_t$, which could be done by making $p_t^{\hat{\pi}}(x_t^\pi) \geq \max\{c_t, p_t^\pi(x_t^\pi)\}$.

Formally, the policy is given as follows:

**Theorem 6.2.** *For any revenue sharing scheme $\pi$, the corresponding revenue sharing scheme $\hat{\pi}$ defined by Algorithm 2 satisfies constraints* (6) *and* (7) *in every sample path.*

## 6.3 Hybrid Revenue Sharing Policy

In the next section we will evaluate the policies discussed on data from a major ad exchange. One conclusion will be that while the prefix policy satisfies more stringent business constraints, it had poor

---
**ALGORITHM 2:** Converting any revenue sharing scheme to the one that obeys constraints (6) and (7).
---
1: For any give revenue sharing scheme $\langle r_t^\pi, p_t^\pi \rangle$.
2: Let $B \leftarrow 0$ be the bank account of the seller.
3: **for** $t = 1, \ldots, T$ **do**
4:     Set the reserve price $r_t^{\hat{\pi}} = r_t^\pi$
5:     **if** item is sold, that is, $b_t^f \geq r_t^{\hat{\pi}}$ **then**
6:         Collect the buyers' payment $x_t^{\hat{\pi}} = x_t^\pi$
7:         Pay the seller $p_t^{\hat{\pi}}(x_t^{\hat{\pi}}) = \max\left\{c_t, (1-\alpha)x_t^{\hat{\pi}} - B, p_t^\pi(x_t^{\hat{\pi}})\right\}$
8:         Update the bank account $B \leftarrow B + p_t^{\hat{\pi}}(x_t^{\hat{\pi}}) - (1-\alpha)x_t^{\hat{\pi}}$
9:     **end if**
10: **end for**
---

performance in terms of exchange profit compared, for example, with the refund policy. Our goal in this section is to combine the insights on optimal formats of reserve prices and revenue sharing policies from the theory in Sections 3 and 4 together with empirical observations from experiments. Thus motivated, we design a hybrid policy that has profit performance compared with the refund policy but satisfies the stringent constraints in the previous section.

---
**ALGORITHM 3:** Hybrid multi period prefix policy.
---
1: Let $B \leftarrow 0$ be the bank account of the seller.
2: Determine the optimal dual variable $\mu^* \in \arg\min_{0 \leq \mu \leq 1} \hat{\phi}(\mu)$
3: **for** $t = 1, \ldots, T$ **do**
4:     Set the reserve price $r_t^\pi = \max\{\min\{\bar{c}_t, r^*(c_t(\mu^*))\}, r^*(0)\}$
5:     **if** item is sold, that is, $b_t^f \geq r_t^\pi$ **then**
6:         Collect the buyers' payment $x_t^\pi = \max(r_t^\pi, b_t^s)$
7:         Pay the seller $p_t^\pi(x_t^\pi) = \max\left\{c_t, (1-\alpha)x_t^{\hat{\pi}} - B\right\}$
8:         Update the bank account $B \leftarrow B + p_t^\pi(x_t^\pi) - (1-\alpha)x_t^\pi$
9:     **end if**
10: **end for**
---

We call it a hybrid policy since the reserve price $r_t^\pi$ is a hybrid of the reserve prices computed in Sections 3 and 4. The payment to sellers is the least payment required to satisfy the prefix revenue share constraint (6) and the per-period opportunity cost constraint (7).

# 7 Empirical Evaluation

In this section, we use anonymized real bid data from a major ad exchange to evaluate the policies we discussed in previous sections. Our goal will be to validate our insights on data. In the theoretical part of this paper we made simplifying assumptions, that not necessarily hold on data. For example, we assume quasi-concavity of the expected profit function $\Pi(r, c)$. Even though this function is not concave, we can still estimate it from data and optimize using linear search. Our theoretical results also assume we have access to distributions of buyers' bids. We build such distributions from past data. Finally, in our real data set bids are not necessarily stationary and identically distributed over time. Even though there might be inaccuracies from bids changing from one day to another, our revenue sharing policies are also robust to such non-stationarity.

## 7.1 Data Sets

The data set is a collection of auction records, where each record corresponds to a real time auction for an impression and consists of:

- a seller (publisher) id,
- the seller declared opportunity cost,
- a set of bid records. Each bid record corresponds to a buyer id and the value of the bid submitted by that buyers to the auction.

The maximum revenue share $\alpha$ that the intermediary could take is set to be a constant. To show that our results do not rely on the selection of this constant, we run the simulation for different values of $\alpha$ ($\alpha = 0.15, 0.2, 0.25$).

Our data set will consist of a random sample of auctions from 20 large publishers over the period of 2 days. We will partition the data set in a *training set* consisting of data for the first day and a *testing set* consisting of data for the second day.

## 7.2 Preprocessing Steps

Before running the simulation, we need to do some preprocessing of the data set. The goal of the preprocessing is to learn the parameters required by the policies we introduced for each seller, in particular, the optimal reserve function $r^*$ and the optimal Lagrange multiplier $\mu^*$. We will do this estimation using the training set, i.e., the data from the first day.

The first problem is to estimate $\Pi(r, c)$ and $r^*(c)$. In order to estimate $\Pi(r, c)$ for a given impression we look at all impressions in the training set with the same seller and obtain a list of $(b^{\mathsf{f}}, b^{\mathsf{s}})$ pairs. We build the empirical distribution where each of those pairs is picked with equal probability. This allows us to evaluate and optimize $\Pi(r, c)$ with a single pass over the data using the technique described in Paes Leme et al. [2016].

For each seller, to estimate $\mu^*$, we enumerate different $\mu$'s from the discretization of $[0, 1]$ (denoted by $D$) and evaluate the profits of these policies on the training set. Then the estimation ($\hat{\mu}^*$) of $\mu^*$ is the $\mu$ that yields the maximum profit on the training set, i.e.,

$$\hat{\mu}^* \triangleq \arg\max_{\mu \in D} \hat{\mathsf{profit}}(\mu)$$

## 7.3 Evaluating Revenue Sharing Policies

We will evaluate the different policies discussed in the paper on testing set (day 2 of the data set) using the parameters $\hat{r}^*(c)$ and $\hat{\mu}^*$ learned from the training set during preprocessing. For each revenue sharing policy we evaluate, we will be concerned with the following metrics: *profit* of the exchange, *payout* to the sellers, *match rate* which corresponds the number of impressions allocated, *revenue* extracted from buyers and *buyers values* which is the sum of highest bids over allocated impressions (here we assume that buyers report their values truthfully in the second-price auction run by the exchange). In addition, the average *intermediary's revenue share* will be calculated.

The policies evaluated will be the following:

- `NAIVE`: naïve policy (Section 2),
- `SINGLE`: single period policy (Section 3),
- `REFUND`: multi period refund policy (Algorithm 1 in Section 6.1),
- `PREFIX`: multi period prefix policy (Algorithm 2 in Section 6.2),
- `HYBRID`: multi period hybrid policy (Algorithm 3 in Section 6.3).

In Table 1, we report the results of the policies for different values of $\alpha$ (0.15, 0.2, 0.25). The metrics are reported with respect to the `NAIVE` policy. In other words, the cell in the table corresponding to revenue of policy `P` is the revenue lift of `P` with respect to `NAIVE`:

$$\mathsf{revenue\ lift}(\mathtt{P}) = \frac{\mathsf{revenue}(\mathtt{P})}{\mathsf{revenue}(\mathtt{NAIVE})} - 1$$

The only metric that is not reported as a percentage lift is the revenue share in the last column which corresponds to:

$$\mathsf{rev\ share}(\mathtt{P}) = \frac{\mathsf{profit}(\mathtt{P})}{\mathsf{revenue}(\mathtt{P})}$$

## 7.4 Interpreting Simulation Results

What conclusions can we draw from the lift numbers? The first conclusion is that even though the theoretical model deviates from practice in a number of different ways (concavity of $\Pi(r, c)$, precise

(a) $\alpha = 0.15$

| policy | profit | payout | match rate | revenue | buyers values | rev. share |
|---|---|---|---|---|---|---|
| NAIVE | 0.00% | 0.00% | 0.00% | 0.00% | 0.00% | 15.00% |
| SINGLE | +1.23% | +1.74% | +0.83% | +1.66% | +0.83% | 14.94% |
| REFUND | +8.53% | +8.53% | +3.71% | +8.53% | +7.81% | 15.00% |
| PREFIX | −3.60% | −1.97% | −23.96% | −2.22% | −6.40% | 14.79% |
| HYBRID | +3.34% | +5.38% | +4.09% | +5.08% | +3.31% | 14.75% |

(b) $\alpha = 0.20$

| policy | profit | payout | match rate | revenue | buyers values | rev. share |
|---|---|---|---|---|---|---|
| NAIVE | 0.00% | 0.00% | 0.00% | 0.00% | 0.00% | 20.00% |
| SINGLE | +1.29% | +2.33% | +0.86% | +2.12% | +1.11% | 19.84% |
| REFUND | +9.37% | +9.37% | +8.30% | +9.37% | +9.09% | 20.00% |
| PREFIX | −2.17% | +0.69% | −21.87% | +0.12% | −4.41% | 19.54% |
| HYBRID | +3.81% | +5.93% | +5.26% | +5.51% | +3.78% | 19.68% |

(c) $\alpha = 0.25$

| policy | profit | payout | match rate | revenue | buyers values | rev. share |
|---|---|---|---|---|---|---|
| NAIVE | 0.00% | 0.00% | 0.00% | 0.00% | 0.00% | 25.00% |
| SINGLE | +1.64% | +2.97% | +1.07% | +2.64% | +1.39% | 24.76% |
| REFUND | +9.55% | +9.57% | +10.71% | +9.56% | +9.64% | 25.00% |
| PREFIX | −1.00% | +2.16% | −18.51% | +1.37% | −2.90% | 24.41% |
| HYBRID | +4.61% | +6.90% | +6.74% | +6.33% | +4.55% | 24.60% |

Table 1: Performance of the policies for different $\alpha$'s.

distribution estimates, stationarity of bids), we are still able to improve over the naïve policy. Notice that the naïve policy implements the optimal reserve price subject to a fixed revenue sharing policy. So all the gains from reserve price optimization are already accounted for in our baseline.

We start by observing that even for SINGLE, which is a simple policy, we are able to considerably improve over NAIVE across all performance metrics. This highlights that the observation that "profit and revenue can be improved by reducing the share taken by the exchange" is not only a theoretical possibility, but a reality on real-world data.

Next we compare the lifts of SINGLE, which enforces revenue sharing constraints per impression, versus REFUND, which enforces constraints in aggregate. We can see that the lift is 5.5 to 7 times larger for REFUND compared to SINGLE. For $\alpha = 0.15$, the lift[5] for SINGLE is +1.23% while REFUND is +8.53%. This shows the importance of optimizing revenue shares across all auctions instead of per auction. Additionally, we observe that the match rate and buyers values of REFUND are higher than those of SINGLE. This is in agreement with Proposition 5.1: because the reserve price of the single-period constrained model is typically larger than the one of the multi-period constrained model, we expect REFUND to clear more auctions, which in turns leads to higher buyer values.

Next we analyze the performance of PREFIX and HYBRID policies. While PREFIX is able to raise payout and revenue in some cases, it fails to have a positive impact on profit in all experiments. In PREFIX the exchange ends up sacrificing too much of its revenue share. At first glance, such result seems to be counterintuitive. However, it is not surprising because there is no theoretical guarantee on the profit of policy PREFIX at all. In particular, PREFIX is subject to tighter constraints than REFUND,

and the reserve prices of policy `SINGLE` and policy `NAIVE` are not achievable by policy `PREFIX` with $\mu^* \in [0, 1]$ in general.

This is our motivation for policy `HYBRID`. We address the shortcomings of policy `PREFIX` by granting the intermediary more freedom in picking reserve prices. When $\mu^* = 0$, for example, $r^{\texttt{HYBRID}} = r^{\texttt{SINGLE}}$. As a result, we obtain a policy that is consistently better than `SINGLE`. Even though `HYBRID` is not as good `REFUND` in terms of revenue lift, it satisfied the more stringent constraints defined in Section 6.2, which are not necessarily satisfied by `REFUND`.

One other interesting observation is that the larger the revenue share $\alpha$, the larger the improvement. So the higher revenue share the exchange can negotiate with sellers, the more important it is to invest in sophisticated revenue sharing policies.

To sum up, the policies can be ranked as follows in terms of performance:

$$\texttt{REFUND} \succ \texttt{HYBRID} \succ \texttt{SINGLE} \succ \texttt{NAIVE} \sim \texttt{PREFIX}.$$

### 7.5 Effectiveness of Multi-period Policies

In Proposition 5.2 we provide a theoretical comparison of single-period and multi-period revenue sharing policies and concludes that there are two effects at play: the first is the effect of the Lagrange multiplier $1 - \mu^S$ which increases as the cost grows and the second is the expected expected lift provided by second bids over a rescaled version of the cost $\mathbb{E}[b^{\mathfrak{s}} - \bar{c}]^+$ for $\bar{c} = c/(1-\alpha)$. This effect decreases with $c$. For very low values of $c$ there is not a significant difference between policies due to $\mu^S$ being close to $1$. For large values of $c$, there is again no significant difference since the second bid is rarely above the cost. Our theorems indicate that there is a sweet spot for costs values which makes multi-period policies particularly effective with respect to single-period policies.

To verify this hypothesis experimentally on data we perform the following experiment: we choose a rescaling factor between $0$ and $1.5$ and evaluate the profit obtained by both `SINGLE` and `REFUND` when all the costs are rescaled by that factor. We obtain the result in Figure 1. First we observe that the

|              |              |              |
|:------------:|:------------:|:------------:|
| (a) $\alpha = 0.15$ | (b) $\alpha = 0.20$ | (c) $\alpha = 0.25$ |

Figure 1: We compare profit(`SINGLE`) in red and profit(`REFUND`) in blue for different cost rescaling parameters. The absolute values in the $y$-axis are removed for privacy reasons.

total revenue is decreasing in the cost scaling, since the larger the cost the more constrained the optimizaton problem is. But more interestingly, observe that for very small costs (cost scaling close to zero) there is little difference between `SINGLE` and `REFUND` . The gap grows as the costs increase but as costs become large, the gap again closes and the two policies again produce similar revenue. Interestingly, the actual unscaled costs (rescaling factor equal to 1) are in the sweet spot where `REFUND` is particularly more effective than `SINGLE`.

We next provide some intuition for these results. When the opportunity cost is very low, the revenue constraint binds ($\mu = 1$). Both policies ignore the seller's opportunity cost, price according to the Myerson optimal reserve $r^*(0)$, and pay the seller $(1 - \alpha)$ of the buyers' payments. When the opportunity cost is very high, the floor constraint binds ($\mu = 0$). Both policies internalize the seller's opportunity cost, price according to $r^*(c)$ and pay the seller his opportunity cost. Thus, both policies coincide when the opportunity cost is too low or too high. In the intermediate regime, both constraints are binding. Here, `REFUND` can take advantage of the repeated nature of the auctions to accept a

revenue share lower than $\alpha$ or operate at a loss for some impressions. This grants `REFUND` more freedom in optimizing the auction and extracting more revenue.

## Footnotes

[3]See `https://www.uber.com/info/how-much-do-drivers-with-uber-make/`

[4]While in this paper we focus on the dynamic optimization of revenue sharing schemes when agents report truthfully, it is still an interesting avenue of research to study the broader market design question of designing dynamic revenue sharing schemes while taking into account agents' incentives.

[5]The reader might ask how to interpret lift numbers. The annual revenue of display advertising exchanges is on the order of billions of dollars. At that scale, each 1% lift corresponds to tens of millions of dollars in incremental annual revenue. We emphasize that this lift is in addition to that obtained by reserve price optimization, since NAIVE already captures the gains from setting reserve prices optimally given a simple revenue sharing policy.

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

# A Proof of Results

## A.1 Proof of Theorem 3.1

*Proof.* Constraints (2b) and (2c) readily imply that $p_t^\pi(x) \geq \max(c, (1-\alpha)x)$. Because the exchange is maximizing profits he would like to set the payment to the seller as small as possible, which implies that $p_t^\pi(x) = \max(c, (1-\alpha)x)$. Furthermore, because bids are stationary we can simplify the problem to

$$J^S = T \max_r \underbrace{\mathbb{E}\left[\mathbf{1}\{b^{\mathsf{f}} \geq r\}\left(\max(r, b^{\mathsf{s}}) - \max\left(c, (1-\alpha)\max(r, b^{\mathsf{s}})\right)\right)\right]}_{\triangleq \Pi^S(r)}, \tag{8}$$

where we denote by $\Pi^S(r)$ the objective in (8).

Let $\bar{c} = c/(1-\alpha)$. We next show that the objective $\Pi^S(r)$ can be written as

$$\Pi^S(r) = \begin{cases} \bar{\Pi}^S(r) \triangleq \alpha\Pi(r,0) & \text{if } r \geq \bar{c}, \\ \underline{\Pi}^S(r) \triangleq \Pi(r,c) - \mathbb{E}\left[(1-\alpha)b^{\mathsf{s}} - c\right]^+ & \text{if } r < \bar{c}. \end{cases} \tag{9}$$

When $r \geq \bar{c}$ the payment to the seller when the item is sold is $(1-\alpha)\max(r, b^{\mathsf{s}})$, which implies that $\Pi^S(r) = \alpha\mathbb{E}\left[\mathbf{1}\{b^{\mathsf{f}} \geq r\}\max(r, b^{\mathsf{s}})\right] = \alpha\Pi(r,0)$. When $r \leq \bar{c}$ we can write the objective as follows by adding and subtracting the expected cost $c\mathbb{P}\{b^{\mathsf{f}} \geq r\}$:

$$\Pi^S(r) = \Pi(r,c) - \mathbb{E}\left[\mathbf{1}\{b^{\mathsf{f}} \geq r\}\left(\max\left(c, (1-\alpha)\max(r, b^{\mathsf{s}})\right) - c\right)\right]$$

$$= \Pi(r,c) - \mathbb{E}\left[\mathbf{1}\{b^{\mathsf{f}} \geq r\}\left((1-\alpha)\max(r, b^{\mathsf{s}}) - c\right)^+\right]$$

$$= \Pi(r,c) - \mathbb{E}\left[\mathbf{1}\{b^{\mathsf{f}} \geq r, b^{\mathsf{s}} \geq \bar{c}\}\left((1-\alpha)\max(r, b^{\mathsf{s}}) - c\right)^+\right]$$

$$= \Pi(r,c) - \mathbb{E}\left[\left((1-\alpha)b^{\mathsf{s}} - c\right)^+\right]$$

where the third equality follows because the second term is non-zero when $\max(r, b^{\mathsf{s}}) \geq \bar{c}$ which is equivalent to $b^{\mathsf{s}} \geq \bar{c}$ because $r \leq \bar{c}$, and the last equation follows because $b^{\mathsf{s}} \geq \bar{c}$ implies that (i) $b^{\mathsf{s}} \geq r$ because $r \leq \bar{c}$ and (ii) $b^{\mathsf{f}} \geq r$ since $b^{\mathsf{f}} \geq b^{\mathsf{s}}$.

Note that $r^*(0)$ is the maximizer of $\bar{\Pi}^S(r)$ and $r^*(c)$ is the maximizer of $\underline{\Pi}^S(r)$. We prove the result by considering three cases, that is, whether $\bar{c}$ falls above, within or below the interval $[r^*(0), r^*(c)]$.

**Case 1 ($\bar{c} \geq r^*(c)$).** In this case the optimal reserve price will be shown to be $r^*(c)$. For $r < \bar{c}$ we have that $\Pi^S(r) = \underline{\Pi}^S(r)$ for which the maximizer is $r^*(c)$. This solution is feasible because $r^*(c) \leq \bar{c}$. We need to show that the profit of all reserves $r \geq \bar{c}$ are dominated by that of $r^*(c)$. For any $r \geq \bar{c}$ we have that

$$\Pi^S(r) = \bar{\Pi}^S(r) \leq \bar{\Pi}^S(\bar{c}) = \underline{\Pi}^S(\bar{c}) \leq \underline{\Pi}^S(r^*(c)) = \Pi^S(r^*(c)), \tag{10}$$

where the first equality follows because $r \geq \bar{c}$, the first inequality because $\bar{\Pi}^S(r) = \alpha\Pi(r,0)$ is quasi-concave in $r$ and thus $\bar{\Pi}^S(r)$ is non-increasing when $r \geq r^*(0)$ (to the right of the maximizer) together with the fact that $r \geq \bar{c} \geq r^*(0)$, the second equality because of continuity of the objective function at $\bar{c}$, the second inequality because $r^*(c)$ is the maximizer of $\underline{\Pi}^S(r)$, and the last equality because $r^*(c) \leq \bar{c}$. Therefore $r^*(c)$ is the optimal reserve.

**Case 2 ($r^*(0) \leq \bar{c} \leq r^*(c)$).** In this case the optimal reserve price will be shown to be $\bar{c}$. We first show that for all $r \geq \bar{c}$ the profit is dominated by that of $\bar{c}$. For any $r \geq \bar{c}$ we have that

$$\Pi^S(r) = \bar{\Pi}^S(r) \leq \bar{\Pi}^S(\bar{c}) = \Pi^S(\bar{c}),$$

where the first equality follows because $r \geq \bar{c}$ and the first inequality because $\bar{\Pi}^S(r) = \alpha\Pi(r,0)$ is quasi-concave in $r$ and thus $\bar{\Pi}^S(r)$ is non-increasing when $r \geq r^*(0)$ (to the right of the maximizer) together with the fact that $r \geq \bar{c} \geq r^*(0)$. We next show that for all $r \leq \bar{c}$ the profit is dominated by that of $\bar{c}$. For any $r \leq \bar{c}$ we have that

$$\Pi^S(r) = \underline{\Pi}^S(r) \leq \underline{\Pi}^S(\bar{c}) = \Pi^S(\bar{c}),$$

where the first equality follows because $r \leq \bar{c}$ and the first inequality because $\underline{\Pi}^S(r) = \Pi(r,c) - \mathbb{E}\left[(1-\alpha)b^{\mathsf{s}} - c\right]^+$ is quasi-concave in $r$ and thus $\underline{\Pi}^S(r)$ is non-decreasing when $r \leq r^*(c)$ (to the left of the maximizer) together with the fact that $r \leq \bar{c} \leq r^*(c)$.

**Case 3 ($\bar{c} \leq r^*(0)$).** In this case the optimal reserve price will be shown to be $r^*(0)$. This case follows similarly to case 1 and the proof is omitted. □

## A.2 Proof of Theorem 4.1

*Proof.* Consider the following relaxed version of problem (3) when constraints (3c) and (3b) are imposed in expectation instead of surely.

$$\bar{J}^M \triangleq \max_\pi \sum_{t=1}^T \mathbb{E}\left[\mathbf{1}\{b_t^f \geq r_t^\pi\}(x_t^\pi - p_t^\pi(x_t^\pi))\right] \tag{11a}$$

$$\text{s.t.} \sum_{t=1}^T \mathbb{E}\left[\mathbf{1}\{b_t^f \geq r_t^\pi\}(p_t^\pi(x_t^\pi) - (1-\alpha)x_t^\pi)\right] \geq 0, \tag{11b}$$

$$\sum_{t=1}^T \mathbb{E}\left[\mathbf{1}\{b_t^f \geq r_t^\pi\}(p_t^\pi(x_t^\pi) - c)\right] \geq 0, \tag{11c}$$

$$\text{where } x_t^\pi = \max(r_t^\pi, b_t^s). \tag{11d}$$

Notice that the Lagrange relaxation of problem (11) is equivalent to that of problem (3), which implies that $\bar{J}^M \leq \inf_{0 \leq \mu \leq 1} \hat{\phi}(\mu)$. We prove the result by showing that (i) the proposed policy attains the dual objective and (ii) the proposed policy is primal feasible in (11).

**Step 1 (primal objective).** Let $J^\pi$ be the expected performance of policy $\pi$. The expected performance of the current policy is

$$
\begin{aligned}
J^\pi &= \sum_{t=1}^T \mathbb{E}\left[\mathbf{1}\{b_t^f \geq r_t^\pi\}(x_t^\pi - p_t^\pi(x_t^\pi))\right] \\
&= \sum_{t=1}^T \mathbb{E}\left[\mathbf{1}\{b_t^f \geq r_t^\pi\}((1 - \mu^*(1-\alpha))x_t^\pi - (1 - \mu^*)c)\right] \\
&= T\mathbb{E}\left[\mathbf{1}\{b^f \geq r^*(c(\mu^*))\}((1 - \mu^*(1-\alpha))\max(r^*(c(\mu^*)), b^s) - (1-\mu^*)c)\right] \\
&= T(1 - \mu^*(1-\alpha))\Pi\left(r^*(c(\mu^*)), c(\mu^*)\right) = \hat{\phi}(\mu^*),
\end{aligned}
$$

where the third equation follows because the policy is stationary and bids are i.i.d.

**Step 1 (primal feasibility).** Let $\hat{\phi}'(\mu)$ be the derivative of the dual objective, which is given by

$$\hat{\phi}'(\mu) = -T\mathbb{E}\left[\mathbf{1}\{b^f \geq r^*(c(\mu))\}((1-\alpha)\max(r^*(c(\mu)), b^s) - c)\right]. \tag{12}$$

The first-order conditions of $\mu^*$ for the dual problem imply that

1. if $\mu^* = 0$ then $\hat{\phi}'(\mu^*) \geq 0$

2. if $\mu^* \in (0, 1)$ then $\hat{\phi}'(\mu^*) = 0$

3. if $\mu^* = 1$ then $\hat{\phi}'(\mu^*) \leq 0$

Let LHS(11c)$^\pi$ be the expectation on the left hand side of the floor constraint (11c) under policy $\pi$. We have that

$$
\begin{aligned}
\text{LHS(11c)}^\pi &= \sum_{t=1}^T \mathbb{E}\left[\mathbf{1}\{b_t^f \geq r_t^\pi\}(p_t^\pi(x_t^\pi) - c)\right] \\
&= \mu^* T\mathbb{E}\left[\mathbf{1}\{b^f \geq r^*(c(\mu^*))\}((1-\alpha)\max(r^*(c(\mu^*)), b_t^s) - c)\right] \\
&= -\mu^*\hat{\phi}'(\mu^*) \geq 0
\end{aligned}
$$

where the second equality follows because $p_t^\pi(x) = (1-\mu^*)c + \mu^*(1-\alpha)x$, third equality follows from the formula for the derivative of the dual objective in (12) and the last inequality from the first-order conditions of $\mu^*$ for the dual problem. Let LHS(11b)$^\pi$ be the expectation on the left hand side of the floor constraint (11b) under policy $\pi$. We have that

$$
\begin{aligned}
\text{LHS(11b)}^\pi &= \sum_{t=1}^T \mathbb{E}\left[\mathbf{1}\{b_t^f \geq r_t^\pi\}(p_t^\pi(x_t^\pi) - (1-\alpha)x_t^\pi)\right] \\
&= -(1 - \mu^*)T\mathbb{E}\left[\mathbf{1}\{b^f \geq r^*(c(\mu^*))\}((1-\alpha)\max(r^*(c(\mu^*)), b_t^s) - c)\right] \\
&= (1 - \mu^*)\hat{\phi}'(\mu^*) \geq 0
\end{aligned}
$$

where the second equality follows because $p_t^\pi(x) = (1 - \mu^*)c + \mu^*(1 - \alpha)x$, the third equality follows from the formula for the derivative of the dual objective and the last inequality from the first-order conditions of $\mu^*$ for the dual problem. □

## A.3 Proof of Theorem 4.3

*Proof.* We will again apply the Lagrangian relaxation technique and derive from it an optimal policy for the problem where the constraints (5) and (3b) are imposed in expectation instead of almost surely. We rewrite the dual function $\phi(\mu, \lambda)$ for the random opportunity cost case as follows,

$$
\varphi(\mu, \lambda) \triangleq \sup_\pi \sum_{t=1}^T \mathbb{E}\left[\mathbf{1}\{b_t^{\mathrm{f}} \geq r_t^\pi\}\left(x_t^\pi - p_t^\pi(x_t^\pi) + \lambda\big(p_t^\pi(x_t^\pi) - c_t\big) + \mu\big(p_t^\pi(x_t^\pi) - (1-\alpha)x_t^\pi\big)\right)\right]
$$

$$
= T\big(1 - \mu(1-\alpha)\big)\sup_{r(c)}\mathbb{E}_c\left[\Pi\left(r(c), c(\mu)\right)\right] + \mathcal{X}_{\{\lambda + \mu = 1\}}.
$$

Again, to prevent the last term $\mathcal{X}_{\{\lambda + \mu = 1\}}$ being unbounded, $\lambda + \mu = 1$ and then the optimal dual objective is given by

$$
\inf_{\mu \geq 0, \lambda \geq 0: \mu + \lambda = 1} \varphi(\mu, \lambda) = \inf_{0 \leq \mu \leq 1} \varphi(\mu, 1 - \mu) = \inf_{0 \leq \mu \leq 1} \hat{\varphi}(\mu),
$$

where

$$
\hat{\varphi}(\mu) \triangleq \phi(\mu, 1 - \mu) = T\big(1 - \mu(1-\alpha)\big)\sup_{r(c)}\mathbb{E}_c\left[\Pi\left(r, c(\mu)\right)\right].
$$

Because the reserve price can be adjusted depending on the cost (i.e., the reserve price is measurable w.r.t. the publisher's opportunity cost), we can interchange the order of the supreme $\sup$ and expectation $\mathbb{E}$ to obtain that

$$
\hat{\varphi}(\mu) = T\big(1 - \mu(1-\alpha)\big)\mathbb{E}_c\left[\sup_r \Pi\left(r, c(\mu)\right)\right] = T\big(1 - \mu(1-\alpha)\big)\mathbb{E}_c\left[\Pi\left(r^*(c(\mu)), c(\mu)\right)\right].
$$

We omit the rest of proof as it follows the same steps as in the proof of Theorem 4.1 except that all the expectations are now taken over $c$ as well. □

## A.4 Proof of Proposition 5.1

*Proof.* We prove the result by considering three cases, that is, whether $\bar{c}$ falls above, within or below the interval $[r^*(0), r^*(c)]$.

**Case 1 ($\bar{c} \geq r^*(c)$).** In this case $r^S = r^*(c)$. The result follows because

$$
r^M = r^*(c(\mu^*)) \leq r^*(c(0)) = r^*(c) = r^S,
$$

where the inequality follows because $\mu^* \geq 0$, $r^*(\cdot)$ is non-decreasing and $c(\cdot)$ is non-increasing.

**Case 2 ($r^*(0) \leq \bar{c} \leq r^*(c)$).** In this case $r^S = \bar{c}$. First note that $c(\mu) = \frac{(1-\mu)c}{1-\mu(1-\alpha)}$ is non-increasing in $\mu$, $c(0) = c$ and $c(1) = 0$. Thus there exist $\mu^S \in [0, 1]$ such that $r^*(c(\mu^S)) = r^S$. We claim that $\hat{\phi}'(\mu^S) \leq 0$, which implies that $\mu^* \geq \mu^S$ and as a result $r^M = r^*(c(\mu^*)) \leq r^*(c(\mu^S)) = r^S$ because $r^*(\cdot)$ is non-decreasing and $c(\cdot)$ is non-increasing.

We prove the claim that $\hat{\phi}'(\mu^S) \leq 0$. Because $r^*(c(\mu^S)) = r^S$ and using the formula for $\hat{\phi}'(\cdot)$ in (12) we have that

$$
\hat{\phi}'(\mu^S) = -T\mathbb{E}\left[\mathbf{1}\{b^{\mathrm{f}} \geq r^S\}\left((1-\alpha)\max(r^S, b^{\mathrm{s}}) - c\right)\right]
$$

$$
= -T\mathbb{E}\left[\mathbf{1}\{b^{\mathrm{f}} \geq r^S\}\left(\max(r^S, b^{\mathrm{s}}) - c\right)\right] + \alpha T\mathbb{E}\left[\mathbf{1}\{b^{\mathrm{f}} \geq r^S\}\max(r^S, b^{\mathrm{s}})\right]
$$

$$
= \alpha T\Pi(r^S, 0) - T\Pi(r^S, c) = T\bar{\Pi}^S(r^S) - T\underline{\Pi}^S(r^S) - T\mathbb{E}\left[(1-\alpha)b^{\mathrm{s}} - c\right]^+, \quad (13)
$$

where the last equation follows from the fourth equation from (9). Therefore because $r^S = \bar{c}$

$$
\hat{\phi}'(\mu^S) = T\bar{\Pi}^S(\bar{c}) - T\underline{\Pi}^S(\bar{c}) - T\mathbb{E}\left[(1-\alpha)b^{\mathrm{s}} - c\right]^+ = -T\mathbb{E}\left[(1-\alpha)b^{\mathrm{s}} - c\right]^+ \leq 0
$$

where the last equation follows because $\bar{\Pi}^S(\bar{c}) = \underline{\Pi}^S(\bar{c})$ since the objective of the single period constrained model is continuous at $\bar{c}$.

**Case 3 ($\bar{c} \leq r^*(0)$).** In this case $r^S = r^*(0)$. From case 3 of the proof of Proposition 5.2 we have that $J^M = J^S$ which implies that $r^S$ is an optimal solution for the multi-period constrained model and $r^S = r^M$. □

## A.5 Proof of Proposition 5.2

*Proof.* The first inequality $J^S \leq J^M$ is trivial because every policy of the single period constrained problem is feasible in the multi period constrained problem. For the second bound, we prove the second result by comparing the optimal objective value of the single period constrained problem to the objective value of the Lagrange relaxation.

First note that $c(\mu) = \frac{(1-\mu)c}{1-\mu(1-\alpha)}$ is non-increasing in $\mu$, $c(0) = c$ and $c(1) = 0$. Thus there exist $\mu^S \in [0,1]$ such that $r^*(c(\mu^S)) = r^S$. As a result:

$$
\begin{aligned}
J^M &\leq \inf_{0 \leq \mu \leq 1} \hat{\phi}(\mu) \leq \hat{\phi}(\mu^S) = T\big(1 - \mu^S(1-\alpha)\big)\Pi\left(r^*(c(\mu^S)), c(\mu^S)\right)\\
&= T\mathbb{E}\left[\mathbf{1}\{b^{\mathrm{f}} \geq r^S\}\left(\big(1 - \mu^S(1-\alpha)\big)\max(r^S, b_t^{\mathrm{s}}) - (1 - \mu^S)c\right)\right]\\
&= \mu^S \alpha T\Pi(r^S, 0) + (1 - \mu^S)T\Pi(r^S, c)\\
&= \mu^S T\bar{\Pi}^S(r^S) + (1 - \mu^S)T\underline{\Pi}^S(r^S) + (1 - \mu^S)T\mathbb{E}\left[(1-\alpha)b^{\mathrm{s}} - c\right]^+ ,
\end{aligned}
\tag{14}
$$

where the first inequality follows from weak duality; the second inequality because $\mu^S \in [0,1]$ is dual feasible; the first equality follows from (4); the second equality follows because $r^*(c(\mu^S)) = r^S$ together with $c(\mu) = \frac{(1-\mu)c}{1-\mu(1-\alpha)}$; and the fourth equation from (9). We conclude the proof by considering three cases, that is, whether $\bar{c}$ falls above, within or below the interval $[r^*(0), r^*(c)]$.

**Case 1 ($\bar{c} \geq r^*(c)$).** In this case $r^S = r^*(c)$, which implies that $\mu^S = 0$ because $c(0) = c$ where $c(\mu) = \frac{(1-\mu)c}{1-\mu(1-\alpha)}$. Here (14) gives

$$
J^M \leq T\underline{\Pi}^S(r^*(c)) + T\mathbb{E}\left[(1-\alpha)b^{\mathrm{s}} - c\right]^+ = J^S + \mathbb{E}\left[(1-\alpha)b^{\mathrm{s}} - c\right]^+ ,
$$

where the last equation follows because $T\underline{\Pi}^S(r^*(c)) = J^S$ since the optimal reserve price in the single period constrained model is $r^*(c)$ and $r^*(c) \leq \bar{c}$.

**Case 2 ($r^*(0) \leq \bar{c} \leq r^*(c)$).** In this case $r^S = \bar{c}$ and $\mu^S \in [0,1]$. Here (14) gives

$$
\begin{aligned}
J^M &\leq \mu^S T\bar{\Pi}^S(\bar{c}) + (1 - \mu^S)T\underline{\Pi}^S(\bar{c}) + (1 - \mu^S)T\mathbb{E}\left[(1-\alpha)b^{\mathrm{s}} - c\right]^+\\
&= J^S + (1 - \mu^S)T\mathbb{E}\left[(1-\alpha)b^{\mathrm{s}} - c\right]^+ ,
\end{aligned}
$$

where the last equation follows because $T\bar{\Pi}^S(\bar{c}) = T\underline{\Pi}^S(\bar{c}) = J^S$ since the objective of the single period constrained model is continuous at $\bar{c}$ together with the fact the optimal reserve price in the single period constrained model is $\bar{c}$.

**Case 3 ($\bar{c} \leq r^*(0)$).** In this case $r^S = r^*(0)$, which implies that $\mu^S = 1$ because $c(1) = 0$ where $c(\mu) = \frac{(1-\mu)c}{1-\mu(1-\alpha)}$. Here (14) gives

$$
J^M \leq \alpha T\Pi\left(r^*(0), 0\right) = \bar{\Pi}^S(r^*(0)) = J^S ,
$$

where the last equation follows because the optimal reserve price in the single period constrained model is $r^*(0)$ and $r^*(0) \geq \bar{c}$. $\qquad\square$