[Reviews · NeurIPS 2017]

Reviewer 1



In online display advertising, companies such as Google, Facebook organize a marketplace to connect publishers (sellers) and advertisers (buyers). In this type of auction, the publishers can specify a cost which he/she wants to be paid for showing an ad on his/her page. The auction designer often ads value to the transaction through e.g. targeting etc and hence also wants to benefit from the transaction. Often, this is done through a revenue sharing scheme where the auction designer takes a fraction of the revenue paid by the buyer as profit and passes the rest on to the seller. In this paper, the authors introduce a "dynamic revenue sharing" scheme. The idea here is to adjust the revenue fraction which the auction designer keeps, in a dynamic way, so that his/her revenue is increased and possibly fill rates are increased. The authors propose two different schemes. In the single period revenue sharing scheme, the minimum revenue sharing fraction needs to be met for every transaction - although the revenue share can be increased to optimize long term profits. In the multi-period revenue sharing scheme, the revenue sharing fraction only needs to be met in expectation across a large number of transactions. The authors derive the optimization solution in the single period revenue sharing scheme and a heuristic (but asymptotically optimal) solution in the multi period revenue sharing setting. Finally, the authors describe a comparative analysis and empirical evaluation. Main comments: - The paper is well written and provides a lot of good intuition. - I believe dynamic revenue sharing is a very interesting idea with practical applications. - Line 213: it is not clear to me why in the formula for r^*, we are using \Pi(r,0) rather than \Pi(r,c)? - The paper which is presented as such feel a bit incomplete. Section 6 and section 7 feel a bit rushed and don't provide a lot of insight. Minor comments: - Line 13: missing reference numbers. - Line 49: declared -> declare - Line 141: maybe clarify what the Myerson reserve price is here?

Reviewer 2



This paper studies the problem of sharing revenue between the ad exchange platform and the web publishers in ad exchange auctions. In practice, any revenue sharing mechanism has to satisfy certain properties: The payment to the web publisher should cover the publisher’s cost and the exchange can keep at most an alpha fraction of the revenue. The authors consider the single-stage setting and the multiple stage setting. For the single stage setting they show how to compute the optimal payments to the publisher. For the multiple stage setting, they introduce a dynamic revenue sharing scheme that balances the constraints over multiple periods of time and guarantees them in expectation or high probability. The paper also examine the implications of these results via experiments. I find the paper to be difficult to read at some points. One problem is that the paper has quite a lot of jargon e.g., elastic demand, thick/thin markets, opportunity cost, etc. These make the paper difficult to read for the NIPS audience. Another problem seems to be that the authors have tried to include more results that they can nicely discuss in the paper. This has lead to some sections losing their clarity. For example, section 6.1 or 7 need more details. Overall I like the direction the paper is taking. It is quite natural to allow the constraints to be met with high probability (less so in expectation) over multiple periods of time. So, it is interesting but not surprising to see that in the multi-stage setting optimal policy achieves better utility. The authors formalize the difference between these utilities via upper and lower bounds in proposition 5.2. Let me emphasize that in the multi-stage setting, the policy still appears to be static, it is the constraints that are met to varying degrees dynamically. If this is indeed the case, it would be helpful for the authors to clarify early in the paper and make additional comparisons to when the policy does not have to be static. It appears to me that by allowing a dynamic policy one can get strong concentration bounds for satisfying the publisher’s constraints. This might be the insight behind section 6.1, but there is not enough information provided in the section to verify this point. The paper also includes empirical evaluation of the proposed revenue-sharing policies. However, very little information regarding these experiments are included in the main body of the paper, so it’s hard to get a sense of the significance of the empirical results. Given that the authors have some space left in the paper, I suggest that they include more information about these experiments. This is the main section that connects the results of the paper with data science and machine learning, which is a major interest of the NIPS community, whereas the rest of the results mentioned in the paper do not have a learning flavor. After the rebuttal: It would be great if the authors add more details to the experiments section in the NIPS version of the paper. It would be good to also improve the readability. See the 2nd paragraph of my review for specific points that can be improved.

Reviewer 3



This paper studies marketplaces where there is a single seller, multiple buyers, and multiple units of a single good that are sold sequentially. Moreover, there is an intermediary that sells the item for the seller. The intermediary is allowed to keep some of the revenue, subject to various constraints. The underlying auction is the second-price auction with reserve prices. The intermediary must determine how to set the reserve price in order to maximize its profit. Since this auction format is strategy-proof for the buyers, they assume the buyers bid truthfully. They also make the assumption that the seller will report its costs truthfully. First, the authors study “single period revenue sharing,” where at every time step, the intermediary can keep at most an alpha fraction of the revenue and it must cover the seller’s reported cost if the item is sold. Next, they study “multi period revenue sharing,” where the aggregate profit of the intermediary is at most an alpha fraction of the buyers’ aggregate payment and the costs must be covered in aggregate as well. For both of these settings, the authors determine the optimal reserve price. Theoretically, the authors show that multi period revenue sharing has the potential to outperform single period revenue sharing when there are many bidders (the second-highest bid is high), and the seller’s cost is “neither too high nor too low.” They analyze several other model variants as well. The authors experimentally compare the optimal pricing policies they come up with for the various models with the “naïve policy” which sets the reserve price at c/(1-alpha). Their experiments are on real-world data from an ad exchange. They show that the single period and multi period policies improve over the naïve policy, and the multi period policy outperforms the single period policy. This paper studies what I think is an interesting and seemingly very applicable auction format. It’s well-written and the results are not obvious. I’d like to better understand the authors’ assumption that the seller costs are reported truthfully. The authors write, “Note that the revenue sharing contract guarantees, at least partially, when the constraint binds (which always happens in practice), the goals of the seller and the platform are completely aligned: maximizing profit is the same as maximizing revenue. Thus, sellers have little incentive to misreport their costs.” I think this means that in practice, the cost is generally so low in contrast to the revenue that the revenue sharing constraint dominates the minimum cost constraint. This doesn’t really convince me that the seller has little incentive to misreport her cost as very high to extract more money from the exchange. Also, as far as I understand, this statement detracts from the overall goal of the paper since it says that in practice, it’s always sufficient to maximize revenue, so the intermediary can ignore the sellers and can just run the Myerson auction. The application to Uber’s platform is also a bit confusing to me because Uber sets the price for the passenger and the pay for the driver. So is Uber the intermediary? The drivers seem to have no say in the matter; they do not report their costs. =====After author feedback===== That makes more sense, thanks.